# MEMORY REPRESENTATION IN TRANSFORMER

## ABSTRACT

Transformer-based models have achieved state-of-the-art results in many natural language processing tasks. The self-attention architecture allows transformer to combine information from all elements of a sequence into context-aware representations. However, information about the context is stored mostly in the same element-wise representations. This might limit the processing of properties related to the sequence as a whole more difficult. Adding trainable memory to selectively store local as well as global representations of a sequence is a promising direction to improve the Transformer model. Memory-augmented neural networks (MANNs) extend traditional neural architectures with general-purpose memory for representations. MANNs have demonstrated the capability to learn simple algorithms like Copy or Reverse and can be successfully trained via backpropagation on diverse tasks from question answering to language modeling outperforming RNNs and LSTMs of comparable complexity. In this work, we propose and study few extensions of the Transformer baseline (1) by adding memory tokens to store non-local representations, (2) creating memory bottleneck for the global information, (3) controlling memory update with dedicated layer. We evaluate these memory augmented Transformers and demonstrate that presence of memory positively correlates with the model performance for machine translation and language modelling tasks. Augmentation of pre-trained masked language model with memory tokens shows mixed results for tasks from GLUE benchmark. Visualization of attention patterns over the memory suggest that it improves the model's ability to process a global context.

## 1 INTRODUCTION

Transformers (Vaswani et al., 2017) are extremely successful in a wide range of natural language processing and other tasks. Due to the self-attention mechanism transformer layer can be trained to update a vector representation of every element with information aggregated over the whole sequence. As a result, rich contextual representation for every token is generated at the end of encoding. However, a combination of local and global information in the same vector has its limitations. Distributed storage of global features results in "blurring" and makes it harder to access them. Another well-known deficiency of Transformers is poor scaling of attention span that hurts its applications to long sequences.

In our work, we propose and study a simple technique to augment Transformer with memory representation (*MemTransformer*). We extend the Transformer baseline by adding `[mem]` tokens at the beginning of the input sequence and train the model to see if it is able to use them as universal memory storage. To assess the capacity of proposed memory augmentation, we additionally applied it to a number of other architectures. In the *MemCtrl* model update of `[mem]` tokens is controlled by dedicated Transformer layer. *MemBottleneck* model has removed attention between sequence elements, thus making memory the only channel to access global information about the sequence. We also tested memory augmented BERT (Devlin et al., 2019) and Transformer XL (Dai et al., 2019) models.

Our work lies at the intersection of two research directions Memory-augmented neural networks (MANNs) and Transformers. The history of memory augmentation in neural networks is pretty long. Classic *Long-Short Term Memory* (LSTM) (Hochreiter & Schmidhuber, 1997) can be seen as a simple yet powerful form of fine-grained memory augmentation with a single memory value per LSTM cell and memory control logic implemented by internal learnable gates. Thus, in LSTMs,

computations are heavily intertwined with memory. In contrast to that, memory-augmented neural networks incorporate external-memory, which decouples memory capacity from the number of model parameters. *Neural Turing Machines* (NTMs) (Graves et al., 2014) and *Memory Networks* (Weston et al., 2014) are among the best-knows MANNs that provide powerful random access operations over external memory. Memory Networks (Weston et al., 2014; Sukhbaatar et al., 2015) are trained to iteratively reason by combining sequence representation and embeddings in long-term memory with the help of attention. NTMs, and their successors *Differentiable Neural Computer* (DNC) (Graves et al., 2016) and *Sparse DNC* (Rae et al., 2016) are recurrent neural networks equipped with a content-addressable memory, similar to Memory Networks, but with the additional capability to write to memory over time. The memory is accessed by a controller network, typically an LSTM. The full model is differentiable and can be trained via back-propagation through time (BPTT). There is also a line of work to equip neural networks (typically, LSTMs) with data structures like stacks, lists, or queues (Joulin & Mikolov, 2015; Grefenstette et al., 2015). MANN architectures with a more advanced addressing mechanisms such as address-content separation and multi-step addressing were proposed in (Gulcehre et al., 2016; 2017; Meng & Rumshisky, 2018).

Family of Transformer models have been recently applied to many deep learning tasks and proved to be very powerful for the language modeling tasks. The core element of Transformers is self-attention that allows updating representation for every element with information aggregated over the whole sequence. Self-attention scales as $O(N^2)$ with a sequence length, and as a result, it is severely limited in application to long sequences.

There is a separate line of work dedicated to reducing the computational cost of the transformer attention to $O(N\sqrt{N})$ using sparsity (Child et al., 2019), $O(N \log N)$ with local-sensitive hashing (Kitaev et al., 2020) or even $O(N)$ with low-rank approximations (Wang et al., 2020), kernel-based formulation (Katharopoulos et al., 2020), or sparse attention with randomness (Zaheer et al., 2020).

Several recent approaches try to solve this problem by adding some kinds of memory elements to their architecture. *Transformer-XL* (Dai et al., 2019) adds segment-level recurrence with state reuse, which can be seen as a sort of memory. During training, the hidden state sequence computed for the previous segment is fixed and cached to be reused as an extended context when the model processes the next segment. *Compressive Transformer* (Rae et al., 2019) extends the ideas of Transformer-XL by incorporating the second level of the memory into the architecture. Memory on the second level stores information from the short-term memory of the first level in compressed form. *Memory Layers* (Lample et al., 2019) replace a feed-forward layer with a product key memory layer, that can increase model capacity for a negligible computational cost.

Some transformers introduce different sorts of global representations. Among the most recent architectures with global representations are *Star-Transformer* (Guo et al., 2019), *Longformer* (Beltagy et al., 2020), *Extended Transformer Construction* (ETC) (Ainslie et al., 2020) and its successor *Big Bird* (Zaheer et al., 2020). All these architectures reduce full self-attention to some local or patterned attention and combine it with a sparse global attention bottleneck. For example, Longformer uses selected tokens such as `[CLS]` or tokens for question marks to accumulate and redistribute global information to all other elements of the sequence. Among these, the BigBird-ETC with dedicated "global" tokens is the most similar to our MemTransformer approach.

Our MemTransformer, MemCtrl and MemBottleneck Transformer models can be seen as more general limit cases for this class of models. They have dedicated general purpose `[mem]` tokens that can be used by the model as a placeholders to store and process global or copy of local representations. MemTransformer has full self-attention over the memory+input sequence. In contrast, MemBottleneck has full both-way attention between the input sequence and memory but no attention between sequence tokens.

## 2 MEMORY IN TRANSFORMER

### 2.1 BACKGROUND: TRANSFORMER ARCHITECTURE

The process of calculating single Transformer self-attention layer can be seen as a two-step processing flow (see fig. 1a).

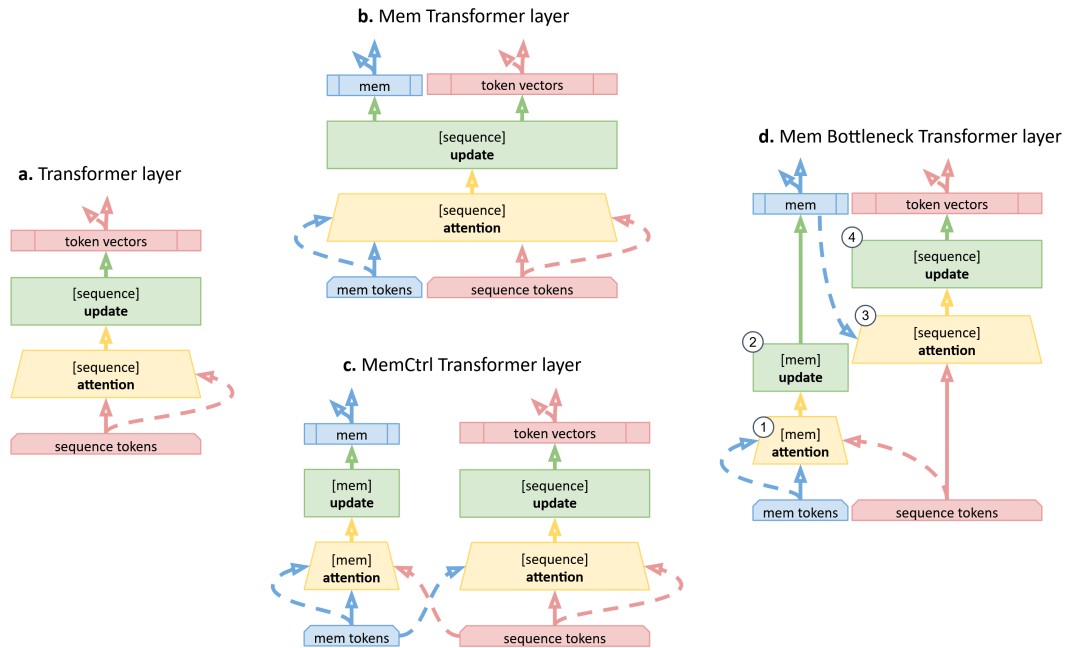

Figure 1: **Memory modifications of Transformer architecture.** (a) *Transformer layer.* For every element of a sequence (solid arrow), self-attention produces aggregate representation from all other elements (dashed arrow). Then this aggregate and the element representations are combined and updated with a fully-connected feed-forward network layer. (b) *Memory Transformer* (Mem-Transformer) prepends input sequence with dedicated [mem] tokens. This extended sequence is processed with a standard Transformer layer without any distinction between [mem] and other elements of the input. (c) Compared to MemTransformer *MemCtrl Transforemer* has dedicated memory controller sub-network. (d) *Memory Bottleneck Transformer* (MemBottleneck Transformer) uses [mem] tokens but separates memory and input attention streams. At the first step, representations of [mem] tokens are updated (2) with the attention span (1) covering both memory and input segments of the sequence. Then representations of input elements are updated (4) with memory attention (3) only. Thus information flow is distributed to representations of elements only through the memory.

1. **Self-attention**. Calculate normalized sum of input $X$ with multi-head attention $MH(Q, K, V)$ between all elements of the sequence:

$$A = LN(X + MH(X, X, X)). \tag{1}$$

2. **Update**. For every element of the sequence update aggregated representation $A$ with $FF$ feed-forward sub-layer then add skip connection and normalize:

$$H = LN(A + FF(A)). \tag{2}$$

## 2.2 SIMPLE MEMTRANSFORMER

The first proposed model is a simple extension of a baseline Transformer we call *MemTransformer*. The idea is to add $m$ special [mem]ory tokens to the standard input (see fig. 1b) then process them in a standard way. So, the input vectors $X$ became the concatenation of the memory token vectors $X^{mem}$ and the original input token vectors $X^{seq}$:

$$X^{mem+seq} = [X^{mem}; X^{seq}] \in \mathbb{R}^{(n+m) \times d}, X^{mem} \in \mathbb{R}^{m \times d}, X^{seq} \in \mathbb{R}^{n \times d}.$$

This modification can be applied independently to encoder and/or decoder. The rest of the Transformer stays the same with the multi-head attention layer processing the extended input.

## 2.3 MEMCTRL TRANSFORMER

In the simple MemTransformer tokens of the memory and the sequence are processed by layers with the same parameters. Thus memory and sequence updated in a similar way. To test if dedicated sub-network for memory update might improve performance we introduce a separate memory control layer (see fig. 1c). Thus, memory representation of *MemCtrl* Transformer is updated as:

$$A^{mem} = LN(X^{mem} + MH^{mem}(X^{mem}, X^{mem+seq}, X^{mem+seq})),$$
$$H^{mem} = LN(A^{mem} + FF^{mem}(A^{mem})).$$

Sequence representation is updated as:

$$A^{seq} = LN(X^{seq} + MH^{seq}(X^{seq}, X^{mem+seq}, X^{mem+seq})),$$
$$H^{seq} = LN(A^{seq} + FF^{seq}(A^{seq})).$$

## 2.4 MEMBOTTLENECK TRANSFORMER

In the MemTransformer input and `[mem]` tokens are updated inside the same traditional self-attend and update processing flow. In this case, representations of the input sequence elements potentially might be updated "as usual" without attending to the content of the memory. Here, global information can propagate in a "peer to peer" manner. To block this distributed information flow and separate storage and processing of global and local representations, we add a memory bottleneck. The resulting *MemBottleneck* Transformer has two-staged processing flow (see fig. 1d).

**1. Memory update.**   First, calculate attention between every memory token and full sequence of memory $X^{mem}$ and input $X^{seq}$ (see Step 1 on the fig. 1d), and update memory token representations (see Step 2 on the fig. 1d):

$$A^{mem} = LN(X^{mem} + MH^{mem}(X^{mem}, X^{mem+seq}, X^{mem+seq})),$$
$$H^{mem} = LN(A^{mem} + FF^{mem}(A^{mem})).$$

**2. Sequence update.**   Calculate attention between sequence and memory (Step 3 on the fig. 1d), and update sequence token representations (Step 4 on the fig. 1d):

$$A^{seq} = LN(X^{seq} + MH^{seq}(X^{seq}, H^{mem}, H^{mem})),$$
$$H^{seq} = LN(A^{seq} + FF^{seq}(A^{seq})).$$

In other words, the memory "attends" to itself and a sequence, and the sequence "attends" only to the memory. This should force the model to accumulate and re-distribute global information through memory. Computations for MemBottleneck Transformer scales linearly with the size of the input sequence or memory $O(NM)$, when the traditional transformer scales as $O(N^2)$.

For all encoder-decoder variants of the memory transformers the decoder part was the same as in the baseline. Output of the last encoder layer $[H^{mem}; H^{seq}]$ passed to the decoder layers.

## 3 RESULTS AND DISCUSSION

As a reference model for a machine translation task we use a vanilla Transformer from official TensorFlow tutorial[1]. Two model sizes were studied for a machine translation task *small*[2] with $N$ = 4 and *base*[3] with $N$ = 6 layers in the encoder. The decoder has the same number of layers as the encoder. For a language modeling task we augmented Transformer XL (Dai et al., 2019) base[4] with 20 `[mem]` tokens. For a masked language model memory augmentation we used pre-trained BERT[5] (Devlin et al., 2019). All values reported in the paper are averaged over 3 runs if otherwise stated.

---

[1] https://www.tensorflow.org/tutorials/text/transformer
[2] $d_{model} = 128, d_{ff} = 512, h = 8, P_{drop} = 0.1, batch = 64, warmup_{steps} = 4000$
[3] $d_{model} = 512, d_{ff} = 2048, h = 8, P_{drop} = 0.1, batch = 64, warmup_{steps} = 32000$
[4] https://github.com/kimiyoung/transformer-xl
[5] `bert-base-cased` checkpoint from HuggingFace Transformers (Wolf et al., 2020) was trained with DeepPavlov (Burtsev et al., 2018) on GLUE tasks.

Table 1: **Performance of baseline and memory models on WMT-14 DE-EN translation.** Values represent an average of BLEU 4 scores for 3 runs of every model evaluated on 2000 samples from WMT-14 DE-EN validation set.

| Small models | | Base models | |
|---|---|---|---|
| 4 layers per encoder/decoder, 20 epochs | | 6 layers per encoder/decoder, 10 epochs | |
| Transformer (baseline) | 19.01 | Transformer (baseline) | 24.65 |
| MemTransformer 5 | **19.17** | - | - |
| MemTransformer 10 | 19.15 | MemTransformer 10 | 25.07 |
| MemTransformer 20 | 19.14 | MemTransformer 20 | 25.58 |
| MemBottleneck Transformer 10 | 11.20 | MemCtrl Transformer 20 | 24.13 |
| MemBottleneck Transformer 20 | 10.41 | MemCtrl Shared Transformer 20 | **25.73** |
| MemBottleneck Skip Transformer 20 | 16.45 | - | - |

## 3.1 PERFORMANCE METRICS

The main hypothesis of the study says that adding memory to multilayered encoder-decoder architectures should result in better performance for sequence processing tasks such as machine translation. BLEU scores for WMT-14 DE-EN translation task (Bojar et al., 2014) are presented in Table 1. After 20 epochs of training, small MemTransformer models have similar scores and clearly outperform the Transformer baseline. Base 6-layer MemTransformer with 10 memory tokens improves the baseline, and doubling the memory up to 20 tokens results in an even higher score of 25.58. This is a modest but solid performance given no hyperparameters fine tuning and beam search were used. MemTransformer results supports our intuition that self-attention could be trained to utilize representations of extra memory elements that are not related to the input sequence to improve the quality of encoding. Surprisingly, adding separate layer for memory control decreases scores below baseline (see MemCtrl 20 in Table 1). On the other hand, memory controller with shared parameters for all 6 encoder layers (MemCtrl Shared 20 in Table 1) demonstrates the best performance among modifications we studied for this task.

The MemTransformer results suggest that if memory extends but not intervene in the Transformer sequence processing, then it is beneficial. But to what extent processing of relations between elements of the sequence can be abstracted to memory? Experiments with MemBottleneck Transformer (Table 1) shows that it is possible, but performance suffers. This can be due to the more complex architecture of the MemBottleneck that has twice more layers in the encoder part (see fig. 1c.). So, it is more difficult and longer to train compared to baseline. On the other hand, degraded performance can also be attributed to the insufficient throughput of the memory bottleneck. Then, there might be a trade-off between the size of the bottleneck and the complexity of learning a deeper network. From the experiments, we see that MemBottleneck 10 learns faster and has lower loss compared to MemBottleneck 20, which points to the complexity of training but not the bottleneck width as a major factor limiting performance.

The limit scenario for the MemBottleneck model is when only memory representations are processed. In MemBottleneck Skip modification of Transformer, representations for sequence tokens are not updated at all (steps 3 and 4 on the fig. 1d are skipped) and encoder output consists of input sequence embeddings and memory representations. Quite unexpectedly, leaving only in memory processing in encoder significantly improves 10.41 BLEU score of MemBottleneck 20 to 16.45 (MemBottleneck Skip in Table 1).

Memory models have better scores after training, but do they require memory for inference? If the performance of trained MemTransformer will stay the same for the inference without `[mem]` tokens, then memory was only needed to improve training and not used for the processing of an input sequence. Results of memory lesion experiments presented in Table 2 demonstrate that removing `[mem]` tokens from MemTransformer input leads to a dramatic drop in BLEU score, from 25.07 to 11.75 for the MemTransformer 10 and from 25.58 to 3.87 for MemTransformer 20 (both models have 6-layers in the encoder). This is an indicator that the presence of `[mem]` tokens is critical for MemTransformer during inference.

Another important question is related to the universality of the learned memory controller. Is it able to utilize memory of arbitrary size, or can work only with the memory capacity it was trained? Memory lesions data (see Table 2) suggest that MemTransformer learns a solution that is partially

Table 2: **Memory lesions.** Performance of the models trained with memory gradually degrades if the memory size is changed during inference.

|  | memory size at inference | | | | | |
|---|---|---|---|---|---|---|
|  | 0 | 2 | 5 | 10 | 20 | 30 |
| MemTransformer 10 | 11.75 | 15.91 | 18.22 | **25.07** | 12.39 | 7.87 |
| MemTransformer 20 | 3.87 | 8.58 | 9.75 | 14.51 | **25.58** | 7.51 |

Table 3: **Memory extension.** Values represent an average of BLEU 4 scores for 3 runs of every model.

|  | 20 epochs | +5 epochs | +10 epochs | +15 epochs |
|---|---|---|---|---|
|  | mem 5 | mem 10 | mem 15 | mem 20 |
| MemTransformer 5 (small) | 19.17 | 19.18 | 19.19 | **19.41** |

Table 4: **Memory augmentation for the language modeling task.** Average performance after training on WikiText-103 (Merity et al., 2016) over 3 runs.

|  | Transformer-XL | + 20 mem fixed pos. emb. | + 20 mem rel. pos. emb. |
|---|---|---|---|
| bpc | 3.182 | 3.179 | **3.176** |
| ppl | 24.09 | 24.02 | **23.95** |

robust to the variations in the size of the memory. BLEU score of MemTransformer 10 with 5 `[mem]` tokens shows it is still able to produce translation that makes sense . On the other hand, if we add 20 more `[mem]` tokens to the same model, it will have scores that are lower even compared to the case when the model is evaluated without memory at all. Interestingly, the model trained with a larger memory size of 20 has weaker generalization abilities. It is evident from the more steep decline of performance with the deviation of memory size from the one used during training.

As we see from memory ablation results (Table 2) increasing memory without fine tuning hurts performance. But, what will happen if the model will be fine-tuned after memory extension? To answer this question we take small MemTransformer 5 pre-trained for 20 epochs and grow it's memory in a few stages up to 20 `[mem]` tokens. On every stage 5 `[mem]` tokens were added and the model was fine tuned for 5 epochs. Results are presented in Table 3. Extension of the memory followed by fine tuning proved to be beneficial and resulted in the model with the highest BLEU score among all small sized modifications.

To test an effect of `mem` tokens on performance in a language modelling task we trained Transformer XL base augmented with memory of size 20. Original Transformer XL has fixed and relative positional encodings, so results for the both options and the baseline are presented in the Table 4. Memory augmentation allows the model to achieve better perplexity.

Positive memory extension results suggested experiments with memory augmentation of an already pre-trained encoders. We took a BERT-base model and augmented it with a memory of different sizes. The model was trained on datasets from the GLUE (Wang et al., 2018) benchmark. Adding `[mem]` tokens to BERT-base model improved its performance on 6 / 9 tasks as shown in the Table 5.

Table 5: **Results on GLUE dev set with `[mem]` tokens added only for end task fine-tuning.** Each `[mem]` was randomly initialized and trained only on the GLUE task. All runs were repeated 5 times and average scores are reported. `+pool` stands for using concatenation of max and avg pooling over outputs for `[mem]` tokens instead of the output from `[CLS]` token for classification.

|  | CoLA | SST-2 | MRPC | STS-B | QQP | MNLI-m/mm | QNLI | RTE |
|---|---|---|---|---|---|---|---|---|
| BERT-base | **62.9** | **92.7** | 90.2/85.8 | 86.0/85.8 | 86.6/89.8 | 83.0/**83.5** | 90.5 | 65.0 |
| 5mem | 61.3 | 92.4 | 90.4/86.4 | 86.0/85.8 | 86.8/90.1 | 82.7/83.3 | 90.7 | **68.0** |
| 5mem+pool | 62.1 | 92.3 | 89.4/84.8 | 85.8/85.6 | 86.9/**90.2** | **83.3**/83.3 | **90.8** | 60.2 |
| 10mem | 60.6 | 92.5 | **91.3/87.6** | 86.6/86.4 | 86.4/89.8 | 82.8/83.3 | 90.5 | 66.8 |
| 10mem+pool | 62.6 | 92.6 | 90.2/86.0 | **86.7/86.5** | 87.1/90.2 | 83.1/83.0 | 90.7 | 61.2 |
| 20mem | 60.9 | 92.4 | 91.2/87.5 | 86.4/86.2 | 86.8/90.1 | 82.8/83.1 | 90.7 | 65.3 |

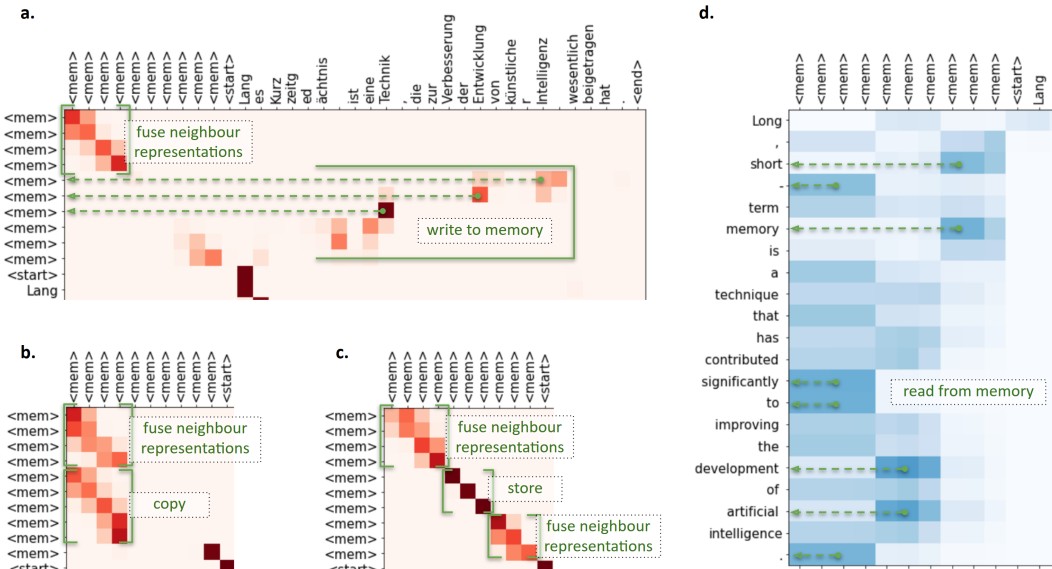

Figure 2: **Operations with memory learned by MemTransformer 10.** (a) The pattern of self-attention in the $3^{rd}$ encoder layer. Here, [mem] tokens in the central segment of memory (on the left) attend to the vector representations of tokens Technik, Entwicklung, Intelligenz (and some others). This attention values are consistent with the *writing* of selected token vectors to the [mem] tokens. Activity in the left top corner that involves first four tokens might indicate *fusion* of neighbour vectors by pairwise summation of [mem] tokens. (b) In the next $4^{th}$ layer of the same encoder similar fusion operation with the same [mem]'s is repeated. A parallel diagonal activity just below the fusion pattern can be attributed to *copy* operation. (c) Another attention head in the same encoder layer demonstrates combination of fusion and *store* operations. Sharp self-attention of three tokens in the middle results in adding vectors to themselves. (d) Attention pattern in decoder layer 4 over the output of $6^{th}$ encoder layer suggest that vectors of [mem] tokens are selectively *read* and added to the output token representations during decoding.

## 3.2 ATTENTION PATTERNS IN MEMORY

Generic system with memory relies on three types of operations, such as writing, reading and processing. In this section we present results of analysis of the inner workings of memory augmented transformers to localize these operations. Following previous studies (Kovaleva et al., 2019; Clark et al., 2019), we visually explored attention patterns. Kovaleva et al. (2019) introduced five categories of self-attention and suggested that only "heterogeneous" patterns that spread attention across all input tokens might extract non-trivial information about the linguistic structure. Numerous observations of attention maps across baseline Transformer and MemTransformer models allow us to conclude that the overall structure and distribution of pattern types in the sequence to sequence part of the attention mechanism are similar. Thus, for further analysis, we skip sequence to sequence attention and focus on memory to sequence, memory to memory, and sequence to memory attention patterns. All attention maps for selected models are presented in the Appendix.

**Memory to sequence attention** makes it possible to selectively update vectors stored in [mem] token positions with representations of input sequence elements. Such an update is a form of soft *write to memory* operation. Indeed, we found many patterns consistent with writing from sequence to memory in all MemTransformer models. One of them is shown in Figure 2a. Write to memory type of attention is more frequent in the first layers and almost absent in the deeper part of the encoder.

**Memory to memory attention** allows recombining vectors of [mem] tokens. We found a few common patterns related to in-memory processing. The most common arrangement of activity is diagonal. The diagonal can be blurred (or "soft"), making local *fusion* of the neighboring memory representations possible. Examples of this operation can be seen in the left top corner of the figures

2a., 2b. and 2c. If diagonal attention is sharp (see fig. 2c. in the middle), then corresponding memory vectors are added to themselves, so their content is amplified and became more error-tolerant. This can be seen as a *store* operation. Another possible operation is a block *copy* (examples can be found in the Appendix). It is usually manifested as a vertical block of attention. In this case, a number of consequent [mem] vectors are updated with the same values aggregated from some another sequence of [mem] vectors. A copy operation can also be performed in a [mem] to [mem] manner with preserving a source order as in figure 2b. or with reversing the order (see Appendix for examples).

**Sequence to memory attention** implements *read from memory* operation and can be found in the first layers of the encoder, but it is more pronounced in the middle layers of the decoder. A typical example of the memory "reading" is presented in Figure 2d. Note, that during decoding token representation is updated by reading from a block of subsequent [mem] tokens.

**The overall pipeline of memory processing** is similar for the different runs and sizes of MemTransformer. It consists of writing some information from the input sequence to the memory in the first layers of the encoder, then followed by memory processing in the intermediate layers and amplification in the output layers of the encoder. During decoding, information is read from memory. Here, the highest "reading" activity is commonly observed in the intermediate decoder layers. Interestingly, memory-related attention patterns usually have a block structure. For example, patterns form the particular MemTransformer 10 presented in Figure 2 suggest that the model had learned to split memory into three blocks. Commonly, the same memory operation is applied to all [mem]s of the same block by one particular head. During memory processing, the model can operate in a block-wise manner, as in Figure 2b, where the "block(1-3)" is copying to the "block(4-6)". We speculate that the block structure of memory processing might reduce error rate because the memory representation is "averaged" over the [mem]s of the block during reading (see fig. 2d). Experiments with MemBottleneck architecture show that the model might be able to learn how to copy representations of the input sequence into the memory of the fixed size and use only this memory during decoding.

## 4 CONCLUSIONS

We proposed and studied a series of memory augmented transformer based architectures *MemTransformer*, *MemCtrl* and *MemBottleneck* transformers. Qualitative analysis of attention patterns produced by the transformer heads trained to solve machine translation task suggests that both models successfully discovered basic operations for memory control. Attention maps show evidence for the presence of memory read/write as well as some in-memory processing operations such as copying and summation.

A comparison of machine translation quality shows that adding general-purpose memory in MemTransformer improves performance over the baseline. Moreover, the final quality positively correlates with the memory size. On the other hand, MemBottleneck Transformer, with all self-attention restricted to the memory only, has significantly lower scores after training.

Memory lesion tests demonstrate that the performance of the pre-trained MemTransformer model critically depends on the presence of memory. Still, the memory controller learned by the model degrades only gradually when memory size is changed during inference. This indicates that the controller has some robustness and ability for generalization. We also found, that extension of memory followed by fine tuning leads to better performance.

Application of proposed technique to language model training as well as fine-tuning of BERT based encoder for a battery of GLUE tasks further demonstrated beneficial effect of memory augmentation. This suggests that simple addition of [mem] tokens can extend almost any "encoder-decoder with attention" framework. It can be also applied to the tasks that depend on the multi-hop reasoning or planning. In this cases memory should help to store and process representations for the intermediate stages of the solution.

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

# A    ATTENTION MAPS FOR MEMORY AUGMENTED TRANSFORMERS

In this section we present attention maps for two representative cases of MemTransformer and Mem-Bottleneck transformer models. For both models we use the same input sequence.

**Input sequence**: *Langes Kurzzeitgedächtnis ist eine Technik, die zur Verbesserung der Entwicklung von künstlicher Intelligenz wesentlich beigetragen hat.*[6]

**Predicted translation MemTransformer 10**: *Long, short-term memory is a technique that has contributed significantly to improving the development of artificial intelligence.*

**Predicted translation MemBottleneck 20**: *The short time memory is a technique that has helped to improve the development of artificial intelligence in a lot of sense.*

**Reference**: *Long-term short-term memory is a technique that has contributed significantly to improving the development of artificial intelligence.*[7]

A short guide to help with interpretation of attention maps is shown on the Figure 3.

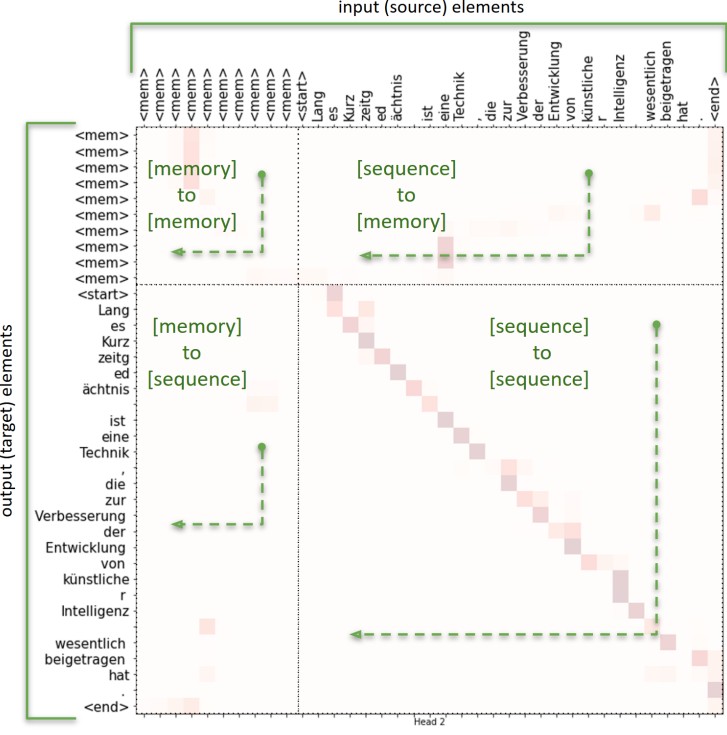

Figure 3: **How to read Memory Transformer attention map.** Attention values indicate how elements of input sequence (on the top) contribute to the update of representation for specific output element (on the left). Attention map for memory augmented transformer can be split into four blocks: (1) *update* - [sequence] to [sequence]; (2) *write* - [sequence] to [memory]; (3) *read* - [memory] to [sequence]; (4) *process* - [memory] to [memory].

## A.1    MEMTRANSFORMER ATTENTION MAPS

Visualisation of attention maps for MemTransformer 10 (see Section 2.2) with a memory size of 10 is presented on the Figure 4 for 6 layers of encoder and on the Figure 5 for 6 layers of decoder. Every transformer layer has 8 attention heads. The model was trained for 10 epochs on WMT-14 DE-EN (Bojar et al., 2014) dataset.

---

[6]https://de.wikipedia.org/wiki/Long_short-term_memory
[7]https://translate.google.com

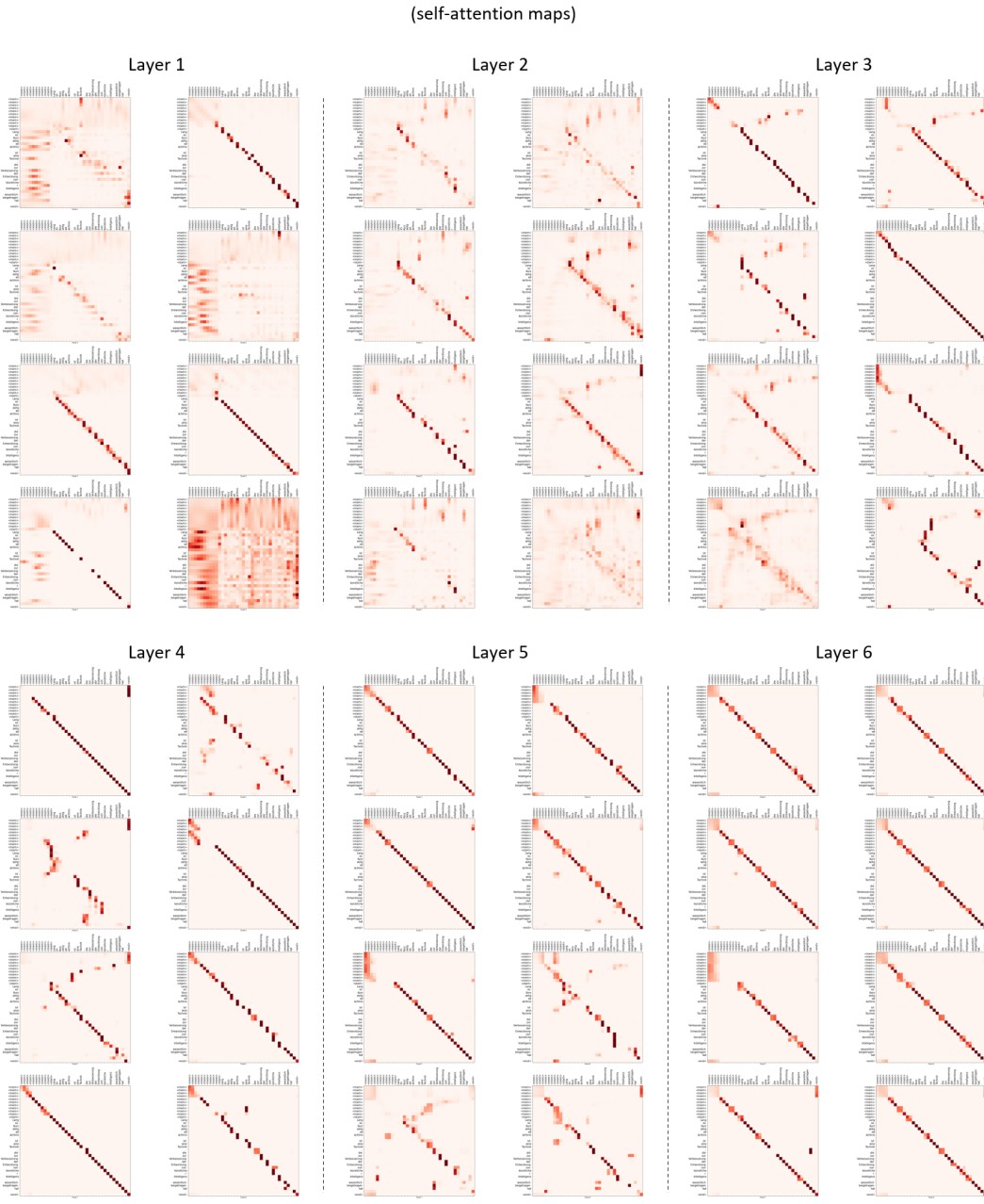

Figure 4: **MemTransformer 10 encoder attention maps.** As the model encodes an input sequence the change of attention patterns related to memory can be interpreted as a *read-process-store* pipeline. Heads in layers 1 to 3 have many *read to memory* patterns. Patterns consistent with *in memory processing* are more frequent in layers 3-6. The last layer is dominated by diagonal attention that can be seen as an amplification of calculated representations.

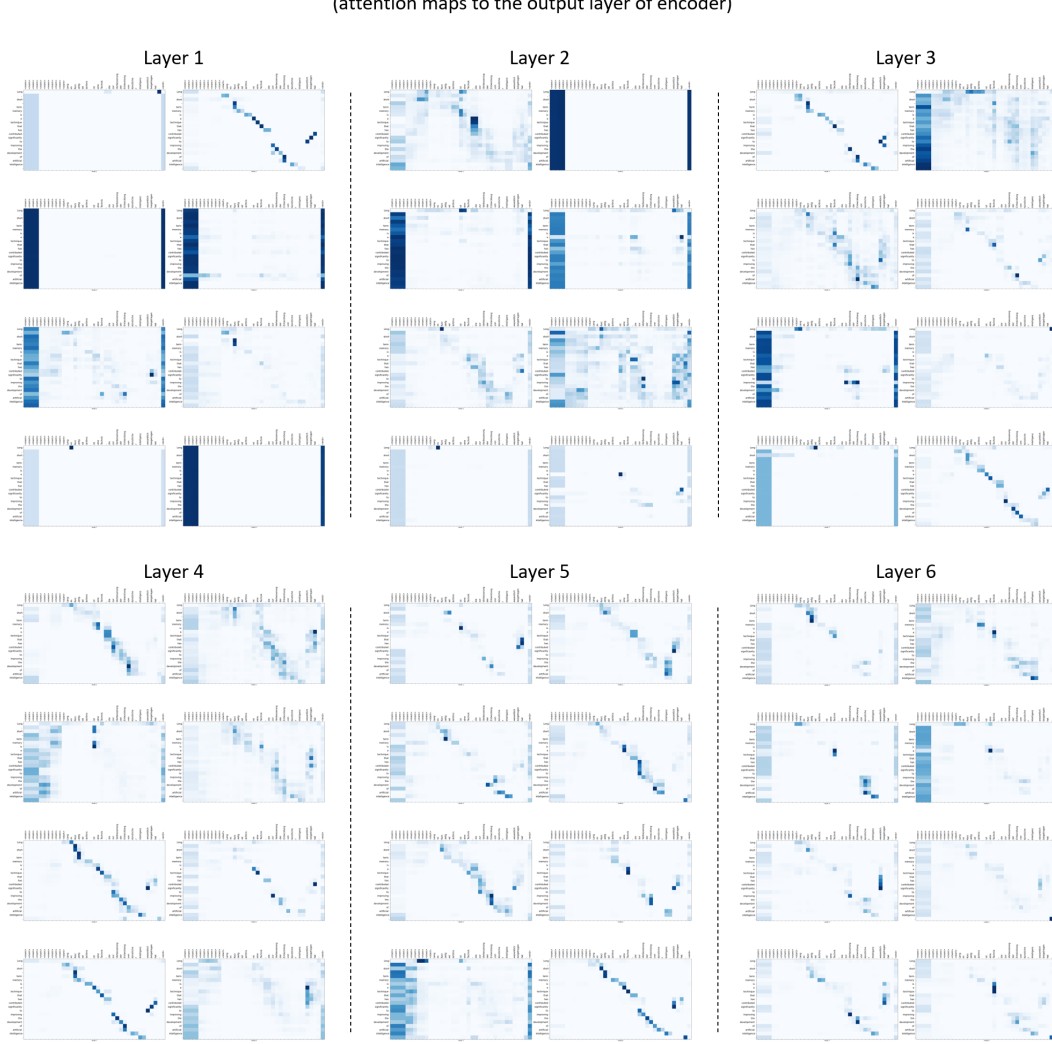

Figure 5: **MemTransformer 10 decoder attention maps.** Every layer of the decoder has heads with signs of memory reading activity. Reading patterns suggest that the representations in memory are locally grouped in 3 blocks.

### A.2 MEMBOTTLENECK TRANSFORMER ATTENTION MAPS

Attention patterns generated by MemBottleneck Transformer architecture (see Section 2.4) strongly suggest that the model learned to copy a given sequence into a memory, process it and use only this representations of input for decoding. The main idea of MemBottleneck is a restriction of global information exchange to memory. Therefore, an update for representations of the input sequence elements can access representations of other elements only by writing into and then reading them from memory. To do that, MemBottleneck uses two different transformer sub-layers each with its' own set of attention heads (see fig. 1c).

Encoder attention maps (see fig. 6) suggest that, as expected, representations for the input elements are copied into memory in layers 1 and 2. Surprisingly, after that they are not properly updated anymore and the decoder mostly attends to the content of memory (see fig. 7). This impressive outcome shows that transformer can be trained to read and process all the information about the input sequence in memory only.

**MemBottleneck Transformer 20 encoder**
(attention maps)

Figure 6: **MemBottleneck 20 encoder attention maps.** In the 1st layer, all attention heads of memory sub-layer ([memory+sequence] to [memory]) read from the input sequence. Only 2 heads of memory sub-layer in the layer 2 reads from the input, but all others are diagonal to amplify content of the memory. No more input reading is present in layers 3 and 4. Notably, all heads of the 1st layer memory attention have patterns that split into three blocks. The top block has sparse attention over the whole sequence without preserving the order. The middle block reads the first half of the sequence in the reverse order, and the bottom block reads the rest in the proper order. This suggests encoding of global information in the top block and local information in the middle and bottom blocks. Layer 3 of memory sub-layer has sharp amplifying diagonals, and something like shifting operations represented by broken diagonals. Layer 4 of memory sub-layer demonstrates mainly heterogeneous patterns which indicates in memory processing. Maps of attention which belongs to the sequence sub-layer ([memory] to [sequence]) of MemBottleneck layer degrade to vertical lines in layers 3 and 4. This is a sing that these attention heads are bypassed as follows from (Kobayashi et al., 2020).

**MemBottleneck Transformer 20 decoder**
(attention maps to the output layer of encoder)

Layer 1

Layer 2

Layer 3

Layer 4

Figure 7: **MemBottleneck 20 decoder attention maps.** At the decoding phase, almost all heads attend to the content of memory but not on the representations of sequence elements.

