# OpenReview forum: "Memory Representation in Transformer"
_ICLR.cc/2021/Conference — Reject_

### Official Review · AnonReviewer4 · 2020-10-27

**Rating:** 3
**Confidence:** 4

**Review:**

This paper proposes to augment transformer architectures with memory components. The high-level idea is to use multiple special “mem” tokens as additional inputs. Depending how the “mem” tokens’ representations interact with the true input sequence, three variants are studied: (a) in MemTransformer [“mem”, input] attends to [“mem”, input]; (b) MemCtrl is similar, but uses a separately parameterized module to calculate the “mem” representations; (c) in MemBottleneck, “mem” attends to [“mem”, input], and [input] attends only to “mem.” Experiments with machine translation, language modeling, and fine-tuning on the GLUE benchmark are conducted.

Overall the presentation of the paper could be significantly improved. For example, it lays out the technical sections in a way that assumes the readers have hands-on expertise with transformer architectures. Further, the motivation is not clear to me. It seems from the experiments that little practical gains can be expected, since MemTransformer never outperforms the transformer baseline in terms of accuracy or efficiency. This is probably fine if the paper studies a set of clear research problems with well-designed experiments and answers some interesting questions. But it is not the case. Last but not least, the experimental setting is flawed: e.g., the MT experiment cuts training at a suspiciously early stage, presumably contributing to the bad performance. In sum I do not think the paper is ready for ICLR.

Pros:
- Studying the transformers through the memory perspective is interesting.

Con:
- Writing can be significantly improved.
- Motivation is not clear: the proposed approach doesn’t seem to bring any interesting practical gain, nor does it answer any interesting research question.
- Experimental design is flawed, making the conclusions less convincing.

Detailed comments:
- I’m not against discussing related works in the introduction. But the current version is probably trying to scramble too much stuff into intro, making it tedious to read.
- I suggest walking through the transformer architecture in a more self-consistent way. I would be very surprised if one without much hands-on experience of transformers can understand what’s going on in section 2.
- The baselines’ performance in the MT experiments is far worse than what we usually see. I’m guessing this is because the training stops too early. Can the authors explain why they choose to do so? If limited computation is the concern, I recommend the authors to work with smaller datasets instead but use a more convincing settings. Same for other experiments.
- The experiments do not show any practical gain from MemTransformer. The paper talked about its linear complexity in input sequence length. This paper would be stronger if it can present convincing experiments and show that MemTransformer can have a better efficiency in practice.
- Section 3.2 would be much more clear and interesting if it opens with the research questions it’s trying to answer.

---

> ### Author Response · Authors · 2020-11-16
> **Thanks for comments.**
>
> > It seems from the experiments that little practical gains can be expected, since MemTransformer never outperforms the transformer baseline in terms of accuracy or efficiency.
>
> This statement contradicts experimental evidence. Results presented in the paper demonstrate that MemTransformer outperforms the baseline for the majority of tasks we tested.
>
> >Motivation is not clear: the proposed approach doesn’t seem to bring any interesting practical gain, nor does it answer any interesting research question.
>
> The motivation is well aligned with a lot of other modern research like Star-Transformer, ETC, Big Bird, Longformer, and others which do use a combination of global and local attention to improve Transformer architecture. Our research also contributes to the area of memory augmented networks, to our knowledge  there are no publications that study transformer modifications in the framework of memory augmentation.
>
> >Last but not least, the experimental setting is flawed: e.g., the MT experiment cuts training at a suspiciously early stage, presumably contributing to the bad performance.
>
> It’s true that the models could be trained longer, and it would probably give us even better results. Yet, 20 epochs do already show a difference, and in terms of the number of updates it is not really small. In general, our results clearly demonstrate that simple augmentation of input with memory tokens improves performance in a variety of tasks from machine translation to classification.
>
> >I’m not against discussing related works in the introduction. But the current version is probably trying to scramble too much stuff into intro, making it tedious to read.
>
> You’re right, it’s worth separating this part into a ‘Related work’ section.
>
> >I suggest walking through the transformer architecture in a more self-consistent way. I would be very surprised if one without much hands-on experience of transformers can understand what’s going on in section 2.
>
> Unfortunately, the size limit of the submission does not allow us to add an introduction to the transformer architecture. We gave a short mathematical formulation of the transformer architecture which is typical for many papers (say, https://arxiv.org/abs/2006.04768 or https://arxiv.org/abs/2003.05997).
>
> >The baselines’ performance in the MT experiments is far worse than what we usually see. I’m guessing this is because the training stops too early. Can the authors explain why they choose to do so? If limited computation is the concern, I recommend the authors to work with smaller datasets instead but use a more convincing settings. Same for other experiments.
>
> Actually, the results on the WMT DE-EN are not ‘far worse than what we usually see’. We should note that we use DE-EN, and not EN-DE dataset. The latter is more common, yet the former is used as well. We chose DE-EN because it is easier to demonstrate that the results are of reasonable quality and to help readers better understand the examples in the paper (just because English is the common ground).  In our experiments 12-layer Transformer (baseline taken from official TF tutorial - https://www.tensorflow.org/tutorials/text/transformer ) achieved 24.65 BLEU after 10 epochs, which is in the top-10 results according to the (see https://paperswithcode.com/sota/machine-translation-on-wmt2014-german-english). To make our results more interpretable, we did not introduce any additional hacks and extensions to the baseline for improving it’s scores.
>
> >The experiments do not show any practical gain from MemTransformer. The paper talked about its linear complexity in input sequence length. This paper would be stronger if it can present convincing experiments and show that MemTransformer can have a better efficiency in practice.
>
> While the results are not SoTA, MemTransformer demonstrates improvements compared to baselines. So, adding [mem] tokens to BERT-base model improved its performance on 6 / 9 tasks as shown in the Table 5. On WMT-14 DE-EN MemTransformer models clearly outperform the Transformer baseline.
>
> >Section 3.2 would be much more clear and interesting if it opens with the research questions it’s trying to answer.
>
> Thank you. We will try to formulate the research question for the Section 3.2 to make interpretation of results easier.

---

> > ### Comment · AnonReviewer4 · 2020-11-18
> > **Thanks for the response!**
> >
> > After reading other reviews as well as the response, I don't find the authors trying to address the central concern shared by several reviewers: the experimental results are far from convincing. The authors argue that the baseline achieves "Top-10" performance with a 24.65 BLEU, but there are only 11 systems on that page, 10 supervised, none autoregressive. If one clicks through any paper on that page, they will probably find an autoregressive transformer baseline (which, in many senses, is the most comparable baseline to this submission), with a 30+ BLEU. With a suspiciously weak baseline, the empirical results are hardly convincing.
> > To be more constructive, I suggest authors train all models for more epochs, so that the transformer baseline matches those by previous works. The current results only show that the proposed approach converges faster.
> >
> > I would keep my initial score unless the authors make serious attempts to address these issues.

---

> > > ### Author Response · Authors · 2020-11-18
> > > **will try to address the common concern.**
> > >
> > > Agree with your point that weak baseline is a common concern among all reviewers. We will try to address it during rebuttal period.

---

### Official Review · AnonReviewer2 · 2020-10-28
**Good qualitative analysis of attention patterns on memory tokens but weaker experimental results**

**Rating:** 5
**Confidence:** 4

**Review:**

The authors propose augmenting transformer architectures by adding trainable memory via introducing special memory tokens to the input sequence. They also explore different architecture extensions (memory bottleneck, dedicated layer for memory updates). The authors’ main hypothesis is that adding memory to Transformer-based architectures should result in better model performance. They present BLEU scores on WMTF-14 DE-EN translation and GLUE benchmark results to support that.

Pros:
+ Good qualitative analysis of attention patterns that emerge when memory tokens are added
+ A practically viable memory extension training scheme that can be used to fine-tune models with added memory tokens

Cons:
- The main claim of the paper is that presence of memory tokens positively correlates with the model performance. At the same time, adding memory tokens increases the model’s capacity, making the comparisons with the baseline unfair. The more rigorous approach would be to compare to a baseline with proportionally increased dimensions as having memory tokens may be akin to having additional space to store sequence token representations. Some of the attention patterns (write-to-memory -> store -> read-from-memory) indicate that this may be the reason for the improved model scores.
- The use of global tokens (like [CLS] token in Longformer as mentioned in the paper) is a well-known technique that undermines the paper’s novelty.
- The authors claim that models’ quality positively correlates with the memory size. However, this statement contradicts the findings in Table 1 and Table 5. The authors briefly discuss the lack of trainability in section 3.1. Still, the authors do not specify conditions under which adding more memory will benefit the model quality.

General comments/questions:
* Memory lesions experiments (table 2). Memory size change during inference seems to lead to drastic drops in model quality. At the same time, the models are not trained to perform in the absence of memory. What if the models are trained on different memory sizes from the beginning with memory size sampled per-batch?
* Memory extension experiments (table 3). Does the model forget how to work with smaller memory sizes during the memory extension process? E.g., what will memory lesions results look like for the memory extension model?
* Are any of the models for the GLUE benchmark trained using a memory extension training scheme?
* It’s not clear if all mem tokens use the same embeddings or if they use separate embeddings (e.g., [mem1], [mem2], etc)
* MemBottleneck results lack comparison with other extensions that lower the computational cost (O(N), O(N log N) extensions, etc)
* Relative positional embeddings seem to work better with memory tokens, are there any more experiments with relative positional embeddings except for the ones in table 5?
* Not all results in Table 1 are symmetrically available for both small and base models. This table lacks the results for MemCtrl Transformer for small models and lacks the results for MemBottleneck Transformer for base models which makes it harder to assess the viability of those architecture extensions.
* No results for MemBottleneck and MemCtrl in table 5. At the moment, both MemBottleneck and MemCtrl look weak; having more results for those extensions may help.
* The authors can add FLOPS/params count to tables 1 and 5 to make a comparison of the models' more convenient.

On rating:

Although extending transformer architectures with memory tokens looks like a practically viable way to improve models’ quality, the experiments are not rigorous enough to confirm this is due to the memory mechanism and not because of the increased model capacity, which can be achieved by a simple model scaling. The well-documented use of global tokens in literature also diminishes the novelty of this paper.

---

> ### Author Response · Authors · 2020-11-16
> **Thanks for insightfull comments and ideas to test.**
>
> >The main claim of the paper is that presence of memory tokens positively correlates with the model performance. At the same time, adding memory tokens increases the model’s capacity, making the comparisons with the baseline unfair. The more rigorous approach would be to compare to a baseline with proportionally increased dimensions as having memory tokens may be akin to having additional space to store sequence token representations.
>
> Thank you for the suggestion. We will perform comparison with the baseline you proposed and include results in the next version of this or other upcoming paper.
>
> >The use of global tokens (like [CLS] token in Longformer as mentioned in the paper) is a well-known technique that undermines the paper’s novelty.
>
> There is a principal difference between [mem] tokens and other service tokens like [CLS]. [CLS] output is directly included in the loss function for the next sentence classification, and tokens like [SEP] directly related to the training task. On the other hand, [mem] tokens are general purpose placeholders for any representation. There is no “target” value for a [mem] token, so its output doesn’t included in the loss function directly.
>
> As well, while [CLS] token is used for a long time, to the best of our knowledge there are no works that perform an analysis of the attention patterns emerged when using specifically this token.
>
> >The authors claim that models’ quality positively correlates with the memory size. However, this statement contradicts the findings in Table 1 and Table 5. The authors briefly discuss the lack of trainability in section 3.1. Still, the authors do not specify conditions under which adding more memory will benefit the model quality.
>
> The statement does not contradict the findings in these tables. MemTransformers consistently produce better results.
>
> >Memory lesions experiments (table 2). Memory size change during inference seems to lead to drastic drops in model quality. At the same time, the models are not trained to perform in the absence of memory. What if the models are trained on different memory sizes from the beginning with memory size sampled per-batch?
>
> Thank you for this idea. We think that such noisy training might increase robustness of the model to the variation in the size of memory during inference.
>
> >Memory extension experiments (table 3). Does the model forget how to work with smaller memory sizes during the memory extension process? E.g., what will memory lesions results look like for the memory extension model?
>
> This is an intersecting research question. It can be quite easily tested without extended experiments. We will include results of such lesioning in the next version of the paper.
>
> >Are any of the models for the GLUE benchmark trained using a memory extension training scheme?
>
> No, but it is a good idea to test.
>
> >It’s not clear if all mem tokens use the same embeddings or if they use separate embeddings (e.g., [mem1], [mem2], etc)
>
> All mem tokens use the same embeddings but positional encoding for every [mem] token was different.
>
> >MemBottleneck results lack comparison with other extensions that lower the computational cost (O(N), O(N log N) extensions, etc)
>
> That’s correct. We did not target lowering the model computational complexity in the paper, so we do not perform comparisons with those models.
>
> >Relative positional embeddings seem to work better with memory tokens, are there any more experiments with relative positional embeddings except for the ones in table 5?
>
> These are the only results using relative positional embeddings. It’s an interesting direction to explore in the future .
>
> >Not all results in Table 1 are symmetrically available for both small and base models. … No results for MemBottleneck and MemCtrl in table 5. At the moment, both MemBottleneck and MemCtrl look weak; having more results for those extensions may help.
>
> True. We did not have a computational budget to perform all the comparisons and less promising MemCtrl and MemBottleneck was dropped from some experiments. However, the experiments conducted give us an understanding of the relative performance improvements/degradations.
>
> >The authors can add FLOPS/params count to tables 1 and 5 to make a comparison of the models' more convenient.
>
> We will add this to the updated version of the paper.

---

### Official Review · AnonReviewer1 · 2020-10-29
**Interesting Idea But Weak Empirical Results**

**Rating:** 4
**Confidence:** 4

**Review:**

The paper presents transformer extensions with global memory.  The basic idea is to augment the input sequence with special memory token [mem].  The authors explore three variants: (a) MemTransformer - just the augmented sequence, (b) MemCtrlTransformer - separate params for memory update, and (c) MemBottleneck - Bottleneck of memory used to update token representations. The authors show gains over vanilla transformer/pretrained baseline for machine translation, language modeling, and the GLUE setup.

I think the idea in general is interesting but it's not particularly novel. The authors cite a lot of recent works in this area, and a particularly similar approach has been explored by Gupta et al in "GMAT: Global Memory Augmentation for Transformers" (authors should cite this work). The main problem with the paper is that the empirical results are quite weak and don't show a consistent trend with regards to utility of memory.

Apart from the machine translation results, it's hard to make a case that the gains on GLUE and language modeling are significant. For the MT results, the baseline results look too weak and the authors don't cite any prior work. Through a quick scan of MT papers, I found papers reporting +6 BLEU scores on the same task and the same architecture (see https://arxiv.org/pdf/1904.09324.pdf).  This is concerning because the baseline model seems highly undertuned.

Among the three variants, the authors only provide results for MemTransformer on language modeling and GLUE.
For machine translation, some of the entries in Table 1 are missing, and the small and base model don't have the same entries.

Other minor comments:
* What do the authors mean by 2000 validation "segments"? Are the segments just sentences? Is it the standard dev set? If not, then why not? If it's the standard dev set, why not refer to it at as WMT-14 DE-EN dev set.
* The authors don't specify the number of attention heads in main text. I presume it's 8, as mentioned in appendix.
* In Table 1, the authors add a result with "MemCtrl Shared Transformer" which they say is the MemCtrl model with shared parameters. Isn't this the same as MemTransformer?
* The authors say in conclusion that "the speed of training positively correlates with the memory size". They don't provide any evidence for this, and if they train for the same number of epochs then increasing memory size should only increase the training time.
* Again in conclusion, the authors say that "the memory controller learned by the model degrades only gradually when memory size is changed during inference." That is clearly not true given the results in Table 2.

Suggested text edits:
* Section 1 - "They have unspecific [mem] tokens that can store global or copy of local information." -> Rephrase this/Expand on this.
* Section 2.2 - "It minimally changes the original model architecture." -> There's no architecture change. Get rid of this line.
* Section 3 - " if the opposite is not stated." -> "if otherwise stated."
* Section 3.2 - "Kovaleva et al. (Kovaleva et al., 2019) " -> Use \citet

---

> ### Author Response · Authors · 2020-11-16
> **Thanks for comments and suggestions.**
>
> >I think the idea in general is interesting but it's not particularly novel. The authors cite a lot of recent works in this area, and a particularly similar approach has been explored by Gupta et al in "GMAT: Global Memory Augmentation for Transformers" (authors should cite this work).
>
> GMAT appeared the same time we published our paper on arxiv, we were not aware of it when working on the paper. We agree, we should mention it in the updated paper. GMAT is clearly related to our approach, yet it is different in almost all the details: attention type, general architecture, set of tasks. GMAT paper also doesn’t perform any analysis of memory patterns.
>
> >The main problem with the paper is that the empirical results are quite weak and don't show a consistent trend with regards to utility of memory.
>
> MemTransformer modifications outperformed baselines in machine translation, language modeling and in 6 out of 9 GLUE tasks. We regard this as a consistent trend for improvement with small but significant gains.
>
> >Apart from the machine translation results, it's hard to make a case that the gains on GLUE and language modeling are significant. For the MT results, the baseline results look too weak and the authors don't cite any prior work. Through a quick scan of MT papers, I found papers reporting +6 BLEU scores on the same task and the same architecture (see https://arxiv.org/pdf/1904.09324.pdf). This is concerning because the baseline model seems highly undertuned.
>
> We do not target the SoTA on MT benchmark as our goal is to estimate the improvements over the baseline. In our experiments 12-layer Transformer (baseline taken from official TF tutorial - https://www.tensorflow.org/tutorials/text/transformer ) achieved 24.65 BLEU after 10 epochs, which is in the top-10 results according to the (see https://paperswithcode.com/sota/machine-translation-on-wmt2014-german-english). To make our results more interpretable, we did not introduce any additional hacks and extensions to the baseline for improving it’s scores.
>
> >Among the three variants, the authors only provide results for MemTransformer on language modeling and GLUE. For machine translation, some of the entries in Table 1 are missing, and the small and base model don't have the same entries.
>
> True. We did not have a computational budget to perform all the comparisons. However, the experiments conducted give us an understanding of the relative performance improvements/degradations.
>
> > What do the authors mean by 2000 validation "segments"? Are the segments just sentences? Is it the standard dev set? If not, then why not? If it's the standard dev set, why not refer to it at as WMT-14 DE-EN dev set.
>
> You are right, “segments” are sentences from standard WMT-14 DE-EN validation set obtained from https://www.tensorflow.org/datasets/catalog/wmt14_translate#wmt14_translatede-en. Corrected in the text.
>
> > The authors don't specify the number of attention heads in main text. I presume it's 8, as mentioned in appendix.
>
> This information is provided in the footnote to the sentence describing the transformer setup at the beginning of the Section 3 (p.4) as a variable $h$.
>
> >In Table 1, the authors add a result with "MemCtrl Shared Transformer" which they say is the MemCtrl model with shared parameters. Isn't this the same as MemTransformer?
>
> In  MemCtrl Shared all 6 layers of the memory control subnetwork have the same parameters. In MemCtrl  every layer of the memory control subnetwork has its own parameters. This is described in the last two sentences of the first paragraph of Section 3.1. (p. 5).
>
> >The authors say in conclusion that "the speed of training positively correlates with the memory size". They don't provide any evidence for this, and if they train for the same number of epochs then increasing memory size should only increase the training time.
>
> Here “training time” was assumed to represent a number of training steps, so the same performance can be achieved faster compared to baseline. This is equivalent to the better quality after training for the same number of steps. We’ve edited this sentence in conclusion to clarify this.
>
> > Again in conclusion, the authors say that "the memory controller learned by the model degrades only gradually when memory size is changed during inference." That is clearly not true given the results in Table 2.
>
> That depends on an interpretation of graduality. The fact is that the stronger deviation in the number of [mem] tokens during the inference leads to worse performance.
>
> >Suggested text edits
>
> Thank you for suggestions, we updated the paper to address them.

---

> > ### Comment · AnonReviewer1 · 2020-11-17
> > **Follow up**
> >
> > > MemTransformer modifications outperformed baselines in machine translation, language modeling and in 6 out of 9 GLUE tasks. We regard this as a consistent trend for improvement with small but significant gains.
> >
> > I would reiterate my earlier point that the proposed models have very weak empirical gains.
> > * See below for my criticism on MT.
> > * The gains for LM are pretty small. I'm sure by just using different random seeds, one of the random seeds can get these gains.
> > * The claim about outperforming in 6 out of 9 GLUE tasks is extremely flawed. For the memory augmented models, the performance is calculated by max-pooling over 5 configurations. So it's not one single configuration that is outperforming on 6 out of 9 tasks. Moreover, I don't see any sensible trends, like increasing the memory leads to gains, etc. Again given the limited gains and inconsistent trends, this feels more like randomness at play than anything else.
> >
> > > We do not target the SoTA on MT benchmark as our goal is to estimate the improvements over the baseline. In our experiments 12-layer Transformer (baseline taken from official TF tutorial - https://www.tensorflow.org/tutorials/text/transformer ) achieved 24.65 BLEU after 10 epochs, which is in the top-10 results according to the (see https://paperswithcode.com/sota/machine-translation-on-wmt2014-german-english). To make our results more interpretable, we did not introduce any additional hacks and extensions to the baseline for improving it’s scores.
> >
> > Firstly, the baseline is just too weak. The baseline is 6-7 BLEU points worse than the baselines reported in other papers. Other reviewers have pointed the same issue. Secondly, you don't get 6-7 BLEU point increase with "hacks", also calling techniques of another area "hacks" is impolite. I'm not asking the baseline to match SOTA but the baseline should at least be competitive with other baselines used in the literature.
> >
> > > True. We did not have a computational budget to perform all the comparisons. However, the experiments conducted give us an understanding of the relative performance improvements/degradations.
> >
> > Sorry but this is quite a lame response. I don't buy that multiple MemTransformer configurations were trained for GLUE but not a single result for other memory architectures.
> >
> > > In MemCtrl Shared all 6 layers of the memory control subnetwork have the same parameters. In MemCtrl every layer of the memory control subnetwork has its own parameters. This is described in the last two sentences of the first paragraph of Section 3.1. (p. 5).
> >
> > My question was how is MemCtrl Shared different from MemTransformer.
> >
> > >  That depends on an interpretation of graduality. The fact is that the stronger deviation in the number of [mem] tokens during the inference leads to worse performance.
> >
> > I'm quite confused. The authors on one hand agree that there's a significant deviation with a change in the number of [mem] tokens, at the same time insist on using "gradually". Sorry, but if the performance drops from 25.07 to 7.87 with a change in memory size then it is not gradual!

---

> > > ### Author Response · Authors · 2020-11-17
> > > **Response**
> > >
> > > >  I'm quite confused. The authors on one hand agree that there's a significant deviation with a change in the number of [mem] tokens, at the same time insist on using "gradually". Sorry, but if the performance drops from 25.07 to 7.87 with a change in memory size then it is not gradual!
> > >
> > > By gradual we mean that decrease in performance correlates with reduction of memory capacity, full 10 [mem] tokens - 25.07, 5 [mem] - 18.22, 2 [mem] - 15.91, 0 [mem] - 11.75. We will reformulate our statement to avoid further possible misunderstanding.
> > >
> > > > See below for my criticism on MT.
> > >
> > > Please, find our opinion on this matter in response to the R5 -
> > > https://openreview.net/forum?id=g1KmTQhOhag&noteId=uJm31kKvY0t

---

> > > > ### Comment · AnonReviewer1 · 2020-11-21
> > > > **Follow Up**
> > > >
> > > > Thanks for the response.
> > > >
> > > > My original criticism, shared by other reviewers, regarding (a) the empirical evidence being weak, and (b) the lack of a complete set of results, still persists.
> > > > So I'm not going to change my score.

---

### Official Review · AnonReviewer3 · 2020-10-30
**Idea is not new and empirical results are not strong.**

**Rating:** 3
**Confidence:** 4

**Review:**

#### Summary:
The paper brought up an interesting limitation of a Transformer network, that information about the context is stored mostly in the same element-wise representation, which might limit the processing of properties related to global information. This work proposed adding memory tokens to store non-local representations and creating memory bottleneck for the global information. The evaluation shows positive results adding memory to a Transformer network.

#### Weakness:
The idea is not entirely new; Memory augmented networks have been proposed in previous work and the paper does not differentiate the method from related work very well.

-The paper is not clear about how to obtain the memory input token. According to the paper, the memory input token seems like a static vector; however, a memory augmented network can retrieve a dynamic vector for global information, with the ability to read and write to the memory.

-The naive Mem Transformer Layer is counter intuitive, as there is little differentiation between the memory layer and the sequence layers. How the memory layer contains additional global information is not explained.
-Figure 1c is confusing, are the arrows crossed or not?

-The formulation of memory representation and sequence representation are exactly the same for the MemCtrl Transformer, which again is very counter intuitive. It is intuitive that we should augment the sequence transformer layer with additional global information, however, we might not want the other way. The memory layers should be updated at coarser granularity.

-The paper is poorly written with many errors in the paper. Please proof read the paper before submission.

#### Detailed feedbacks:
The reviewer finds this paper hard to read as there is not a flow of story in the paper. The layout of the paper and the design of experiments seems very arbitrary.

-The motivation for a memory network like proposed in the paper is not clearly stated. How the memory network can capture long-term dependencies and information is not clearly explained. The design of variants of Memory Transformers seems very arbitrary.

-The paper also lacks a more detailed comparison with related work, like Transformer-XL, Star-Transformer, Longformer, ETC, etc.

-"global"->''global'' (make sure it looks correct in latex).

-".Surprisingly"->". Surprisingly"

-Please improve the quality of writing and ask native speakers to proof read the paper.

---

> ### Author Response · Authors · 2020-11-16
> **Explaining main concepts of the paper**
>
> > The idea is not entirely new; Memory augmented networks have been proposed in previous work and the paper does not differentiate the method from related work very well.
>
> That’s true the idea is not entirely new, yet there is very little research focused on investigating how the memory is utilized in transformers. To the best of our knowledge, no baselines such as proposed in the paper were investigated before. If you know a particular work we should mention, let us know, please.
>
> >The paper is not clear about how to obtain the memory input token.  According to the paper, the memory input token seems like a static vector; however, a memory augmented network can retrieve a dynamic vector for global information, with the ability to read and write to the memory.
>
> In the second paragraph of section 2.2. (bottom of p.3) we introduce and formally define [mem] tokens. We extend vocabulary with extra service [mem] token, which is similar to [SEP] token heavily used in BERT-like models. The [mem] tokens serve as placeholders which allow transformer to process arbitrary vectors dynamically. This is a core idea of the paper and our results demonstrate that after training dynamic updates of these placeholders are consistent with writing, reading and processing operations.
>
> > The naive Mem Transformer Layer is counter intuitive, as there is little differentiation between the memory layer and the sequence layers. How the memory layer contains additional global information is not explained.
>
> In the case of naive Mem Transformer Layer (fig. 1b), there is no differentiation between the memory and sequence layers as described in the Section 2.2 (p.3). Both [mem] tokens and sequence tokens are processed by the same vanilla transformer layers according to eq. (1) and eq. (2) (p.3).
>
> > Figure 1c is confusing, are the arrows crossed or not?
>
> In the case of MemCtrl Transformer, there are two different layers. The first one updates representations of sequence tokens, and the second updates representations of [mem] tokens. Both layers attend to the same concatenation of sequence and memory (sec. 2.3 p.4), this concatenation of attention input is depicted by crossing arrows.
>
> >-The formulation of memory representation and sequence representation are exactly the same for the MemCtrl Transformer, which again is very counter intuitive. It is intuitive that we should augment the sequence transformer layer with additional global information, however, we might not want the other way. The memory layers should be updated at coarser granularity.
>
> Memory representation is the same for all modifications we propose in our paper. The difference lies in the processing of these representations. We are not imposing any specific function on the memory tokens such as global or local or any other representation, the idea is that the transformer should learn memory representation that helps to solve the main task. It is not clear what do you mean by coarser granularity, so we can not address this in our reply.
>
> > -The motivation for a memory network like proposed in the paper is not clearly stated. How the memory network can capture long-term dependencies and information is not clearly explained. The design of variants of Memory Transformers seems very arbitrary.
>
> Disagree. We demonstrate attention patterns and extensively discuss how transformer can accumulate and process information in memory. Our selection of Memory Transformer modifications to study is very straightforward and explained in the first lines of every dedicated section (sections 2.2, 2.3, 2.4). Specifically, the Simple MemTransformer (sec. 2.2. p.3) is the simplest possible memory extension of the baseline transformer. In the MemTransformer memory and sequence are updated by layers with the same parameters but if processing of memory requires specific computation then it is better to train dedicated subnetwork for this. Thus in the MemCtrl we introduce  a separate sublayer for memory processing. In the MemBottleneck we block direct exchange of information between representations of elements of a sequence to test capabilities of the transformer to learn operations for  processing of global information in memory only.
>
> > -The paper also lacks a more detailed comparison with related work, like Transformer-XL, Star-Transformer, Longformer, ETC, etc.
>
> We will add more detailed comparison to these architectures to the final version of the paper.

---

### Official Review · AnonReviewer5 · 2020-11-05
**Review from R5: Interesting exploration but the method and empirical studies are relatively weak**

**Rating:** 4
**Confidence:** 5

**Review:**

> Summary: The paper proposes to study three formulations (MemTransformer, MemCtrl and MemBottleneck) of memory-augmented self-attention transformers, and investigate the influence of adding memory tokens to the model to its overall performance. The authors claim via some experiments on MT and LM that memory augmentation is able to improve the canonical transformer models.

-------------------------

Post-rebuttal thoughts:

While the rebuttal of the authors did clarify some points, I think it is undeniable that the empirical results presented in the current version of the paper are still far from being convincing enough to claim this form of memory augmentation is a useful addition to the Transformers. The visualization of the memory augmentation via attention map is also too handwavy.

While I appreciate the authors' effort to address my questions in the rebuttal phase, I will keep my current score.

-------------------------

**My general opinion**:

I personally think that memory augmentation is a simple but potentially useful technique that explicitly adds memory to a deep network, which, as the authors mentioned in their paper, is not a new topic. Memory augmented networks have been around in many deep models, especially sequence models. However, I find this paper still not strong enough for me to recommend its acceptance in its current form, mainly because 1) the formulations are rather incremental; and 2) the empirical results are too weak to demonstrate (convincingly) the usefulness; and 3) I found some experimental settings uncommon and empirical analysis too handwavy. My detailed question and comments are below.

-------------------------------------

I have some detailed questions/comments.

1. In the MemBottleneck formulation, does the update rule of $A^\text{seq}$ take $X^\text{mem}$ or $H^\text{mem}$? Figure 1d seems to suggest that the memory update at a layer $i$ precedes the update on the sequence tokens. In addition, what is the difference between $H$ and $X$? Are they not the same sequence (i.e., the $H$ from a layer is passed in as $X$ of the next layer)?

2. In the MemBottleneck subsection, the paper claims that its cost "scales linearly with the size of the input sequence $O(N)$" instead of $O(N^2)$. But if the hypothesis of the authors are correct, that these memory modules serve to collect and "re-distribute" the sequence information, shouldn't one generally expect the memory size to be proportionate to the sequence length? For example, I certainly would expect that a sequence of length 8000 would certainly require a different memory size than a sequence length 40. The point is, one cannot simply assume "the size of the memory is constant", just like one cannot assume "the size of the sequence is constant". The more accurate way, for instance, is to say the complexity is $O(NM)$ with $M$ being the memory size.

3. My major concern with this paper is with its various empirical results and experimental settings:
   - i) In machine translation, how exactly does the encoder-decoder self-attention work in the three proposed Mem settings? E.g., does the decoder queries attend to $X^\text{mem}$ only (of the encoder output), or also the $X^\text{seq}$?
   - ii) How does MemCtrl models perform in the small setting of Table 1? (They seem to be slightly better on the larger setting)
   - iii) Overall, I found the numbers reported for WMT'14 de-en in Table 1 **too low** for me to convince of anything. Even though the *much more commonly used setting is WMT'14 en-de*, it should be overall easy to get a >29 BLEU score with a base Transformer model (e.g., [1,2] got >31 for the base Transformer). I am not sure why the authors halted their experiment at 20 epochs and 10 epochs (how many training steps are these though?). Hence, it's not clear to me whether these minor improvements in the below-expectation BLEU scores are truly indicative of the final performance of these models.
    - iv) The paper says no beam search was used. Why? What are the results if you use beam_size = 5?
    - v) The MemBottleneck Skip Transformer is really just a weak form of a typical Transformer, except that you initialized the first layer of the hidden units to 0, and downsample the sequence from the length of $X^\text{seq}$ to the length of $X^\text{mem}$. So it's not a surprise to me that "MemBottleneck Skip Transformer 20" is significantly better than MemBottleneck 10/20.
    - vi) The various ablation studies the authors made in the paper, in my opinion, exactly suggest that the usefulness of these memory augmentations are in doubt. For example, the huge gap between MemBottleneck and canonical transformer, as well as the negative gap between MemBottleneck 10 and MemBottleneck 20, I think is good evidence of this. Another example is the Table 2, where inference time with a larger memory size even degrades the performance. I didn't find a satisfactory explanation from the authors on this.
    - vii) What are the values in Table 3? Are they BLEU scores? Why are they <1?
    - viii) The paper says that "BLEU score of MemTransformer 10 with 5 `[mem]` tokens shows it is still able to translate with acceptable quality". This is false. 7 BLEU score is **A LOT** of difference. If you look at the generation result, they are qualitatively very different.
    - ix) How exactly did you use the memory augmentation for a causal/decoder Transformer for the language modeling task in Table 3? This is not explained in detail in the paper, but how did you handle the potential information leakage problem (i.e., we don't want future tokens to flow back to the past)? If you simply do it in the same way as in encoders of MT by appending it to the sequence, then you can either only *read* from the memory without writing (if you append to the start of the sequence), or only *write* to it without reading (if you append to the end of the sequence). I might be missing something here, and hopefully the authors can clarify.
    - x) The fact that there's no consistent improvement in Table 5 of the memory-augmented Transformer on the base Transformer is very concerning to me. Apparently, sometimes BERT-base is better than many of the MemTransformers.

4. One potentially interesting study that I think the authors can look into is the memory slot permutation. In a certain sense, the memory slots have the same representation embedding `[mem]`, and are mainly differentiated (among themselves) by their respective position embedding. However, it also seems from Figure 2 that when writing to memory, the original order of the sequence tokens is not necessarily preserved. So what if, in the intermediate layers of the MemTransformer, you permute the order of the memory slots? Do you expect that to affect the performance of the MemTransformer?

5. The other issue I found about the paper is that, although this is an empirical paper, there are still a bunch of not well-supported claims (i.e., handwavy claims) which the authors should have looked into. For example:
    - i) In section 3.1: "This can be due to the more complex architecture of the MemBottleneck that has twice more layers in the encoder part". But there's no formal investigation of this in the paper... for example, does an 8-layer MemTransformer (which has "twice as many layers as the 4-layer MemBottleneck) also perform badly?
    - i) In the caption of Figure 2: "Activity in the left top corner that involves first four tokens might indicate fusion of neighbour vectors by pairwise summation of `[mem]` tokens." This sounds like a posterior explanation... and it's not clear to me what this "fusion" is doing and why it's doing this. How is this helping MemTransformer? Does it occur in every head in every layer? Does it happen even if you only use memory size 5? These questions should be answered if you would like to claim this phenomenon.
    - ii) In the "memory to memory attention" paragraph: "If the diagonal attention is sharp, then corresponding memory vectors are added to themselves, so their content is amplified and became more error-tolerant." Do you have support for the error tolerance claim? Is it still "amplified" even if there is a LayerNorm immediately afterward, which would shrink things back anyway (i.e., $\text{LN}(x)$ and $\text{LN}(x+x)$ should be the same)?

-------------------------------------

Some minor points that didn't impact the score:

1. In the intro section, 'results in "blurring"' ---> it's not clear to me what this means. Please clarify.
2. In the very beginning of section 3, the authors used $N=4$ to refer to number of layers. But $N$ has been used to refer to sequence length before (e.g., when saying $O(N)$).

- [1] https://www.ijcai.org/Proceedings/2020/0534.pdf
- [2] https://arxiv.org/pdf/2010.07638v1.pdf

---

> ### Author Response · Authors · 2020-11-16
> **Thanks for in depth review and research questions raised**
>
> > In the MemBottleneck formulation, does the update rule of  $A^{seq}$ take $X^{mem}$ or $H^{mem}$?
>
> You are right that $H^{mem}$  is passed to the next layer as $X^{mem}$ . To make it less confusing we will update equations in Section 2.4.
>
> >The more accurate way, for instance, is to say the complexity is with being the memory size.
>
> Thank you for your suggestions. Your formulation is more accurate and will use it in the paper.
>
> >In machine translation, how exactly does the encoder-decoder self-attention work in the three proposed Mem settings?
>
> We keep the decoder intact, so it attends to concatenation of memory and sequence representations. We will update paper to make this clear.
>
> >How does MemCtrl models perform in the small setting of Table 1?
>
> We compared MemCtrl variants to MemTransformer and baseline only over the base sized models. We expect that MemCtrl with shared weights should have performance similar or better then MemTransformer.
>
> > Overall, I found the numbers reported for WMT'14 de-en in Table 1 too low for me to convince of anything.
>
> You are right that  WMT’14 en-de is more common, but we chose de-en because it is easier to demonstrate that the results are of reasonable quality and to help readers better understand the examples in the paper (just because English is the common ground).  In our experiments 12-layer Transformer (baseline taken from official TF tutorial - https://www.tensorflow.org/tutorials/text/transformer ) achieved 24.65 BLEU after 10 epochs, which is in the top-10 results according to the (see https://paperswithcode.com/sota/machine-translation-on-wmt2014-german-english).
>
>
> > The paper says no beam search was used. Why?
>
> To make our results more interpretable and fair, we did not introduce any additional hacks and extensions for improving scores of the baseline and our code derived from it.
>
> > The MemBottleneck Skip Transformer is really just a weak form of a typical Transformer.. .So it's not a surprise to me that "MemBottleneck Skip Transformer 20" is significantly better than MemBottleneck 10/20.
>
> It is not so obvious for us, because MemBottleneck without skip connection still might learn to pass representations of sequence tokens without modifications through the layers of the encoder.
>
>  >What are the values in Table 3? Are they BLEU scores? Why are they <1?
>
> Values multiplied by 100 will represent BLEU. To be corrected.
>
> >The paper says that "BLEU score of MemTransformer 10 with 5 [mem] tokens shows it is still able to translate with acceptable quality". This is false. 7 BLEU score is A LOT of difference.
>
> We wanted to say that translation still frequently makes some sense (and not just random sequence) even for such low values of BLEU. This will be reformulated.
>
> > How exactly did you use the memory augmentation for a causal/decoder Transformer for the language modeling task in Table 3?
>
> For language modeling we added [mem] tokens after the sequence for fixed positional embeddings and after the sequence for the relative positional embeddings. When [mem] tokens were prepended at the beginning of the sequence they were masked to prevent reading from the future and can be only used to process content of the [mem] and sequence tokens from the previous segment of Transformer XL input. In the case of appending at the end of the sequence, [mem] tokens were able to append for sequence from previous and current segments as well as [mem] tokens from the previous segment.
>
> > The fact that there's no consistent improvement in Table 5 of the memory-augmented Transformer on the base Transformer is very concerning to me.
>
> The BERT memory augmentation differed significantly from other experiments, because memory augmentation was performed for an already pre-trained model. So, it seems that fine tuning for memory usage on the small task is not sufficient to have large enough gains in performance.
>
> >So what if, in the intermediate layers of the MemTransformer, you permute the order of the memory slots?
>
> Fig. 2d suggests that order of memory slots matters because reading operation is performed over a block of nearby [mem] tokens. So, we think that permutation of memory slots in the intermediate layers should disrupt normal memory processing. But this can be only verified by experiments.
>
> >The other issue I found about the paper is that, although this is an empirical paper, there are still a bunch of not well-supported claims (i.e., handwavy claims) which the authors should have looked into.
>
> Majority of such statements are devoted to possible interpretation of phenomena observed in results of experiments. This is why we tried to frame them as discussion but maybe we're not very successful due to incorrect wording. The purpose of that discussion is to draw readers’ attention to some interesting features of the results but not completely explain them. Indeed, many of them require further research to be transformed from conjectures to solid statements.

---

> > ### Comment · AnonReviewer5 · 2020-11-16
> > **Further questions after the author's response**
> >
> > Thank you for providing the explanations to some of the questions. I would like to follow up with some of the responses above:
> >
> > 1) Despite the explanation the authors provided on the experiment settings, I still find the fact that:
> >     - i) MemBottleneck and MemCtrl are not studied or used after Table 1; and
> >     - ii) MemTransformer brings quite minimal empirical improvement (e.g., in LM)
> > make me unconvinced of the benefit of the memory augmentation involved. For instance, in my (lots of) experience with the Transformer-XL model, getting a 23.95 ppl is definitely possible, and if the memory augmentation technique were to be truly useful over long sequences (note: remember that Transformer-XL model uses pretty long sequences for word-level LM, so one probably would expect some "memory help" would matter), I think there should be a larger gap. But the way the paper is currently conducting memory augmentation in language-model Transformers is actually very similar to Transformer-XL. What these "memory slots" are doing, is just downsampling the previous segment of the sequence, and then apply it in the same "caching" way as in Transformer-XL. This does reduce the computation cost by a small amount, but I don't see how it is essentially different.
> >
> > 2)  I still find the WMT14'de-en result not quite satisfactory. Compared to the "10-epoch result", which can be easily affected by the batch size and other hyperparameter settings, it is much more common to compare on a standard number of training steps (e.g., 300K steps). And since it's not a so common practice, it would be more effective to compare the performance on a fair ground without early stopping, e.g., with the [1] and [2] I referred to in my review above. Are the authors able to run more experiments and add the results to the new version of the paper in the rebuttal phase?
> >
> > In addition, the paperswithcode leaderboard for WMT14'de-en only has 9 results (as de-en is indeed uncommon)... so it's not quite the same as getting top 10 in a more competitive benchmark like en-de (https://paperswithcode.com/sota/machine-translation-on-wmt2014-english-german), isn't it? :-) I would suggest the authors running on the en-de benchmark for a future version of the paper, too.
> >
> > Similarly, I think the authors actually **should** "introduce additional hacks and extensions" to their experiments if demonstrating the win in numbers is the most effective way of showcasing usefulness of the approach. I'm not talking about anything fancy like NAT generation or adaptive length penalty, but beam search should be pretty standard in bpe-level NMT systems now. I believe the other reviewers raised a similar point below.

---

> > > ### Author Response · Authors · 2020-11-17
> > > **win in numbers is not ony value for research**
> > >
> > > The primary focus in our work was not on improving machine translation or language models per se but to study how general purpose memory slots might be utilized by Transformer architecture. This is why we not agree that hacks **should** be included. We took the model from the library to minimize implementation bias and minimally changed it. We run the both versions for the same hyperparameters and the same number of training steps then report numbers and discuss what happen with memory representation. It would be nice to improve sota by small modification of baseline but it is not always the case.
> > >
> > > For example, XLNet [3] brings quite minimal empirical improvement over the BERT (Table 1, p.7) and it  is hard to find substantial number of XLNet based solutions on the GLUE and SuperGLUE  leaderboards. Was it worth publishing XLNet paper if it is so weak on GLUE\SuperGLUE?
> > >
> > > Anyway, we will try to reproduce stronger baseline till the end of rebuttal period and update results in the paper.
> > >
> > > [3] https://arxiv.org/pdf/1906.08237.pdf

---

> > > > ### Comment · AnonReviewer5 · 2020-11-17
> > > > **Number is not the only value, but should at least be convincing**
> > > >
> > > > Thank you for your response.
> > > >
> > > > I'm glad that the authors point to the XLNet paper, though I would hardly classify it as "quite minimal empirical performance". For example, in the exact "Table 1, p.7" that the authors point out, the XLNet outperforms BERT *across the board* with a good margin (and doing this on a large scale, not just base scale). Moreover, the highlight of XLNet is also different in the sense that it is an autoregressive model, which behaves differently from the original BERT model. The authors of XLNet did a great job showing that their mechanism (e.g., permutation LM) does bring valuable improvement to the autoregressive model to even outperform the bidirectional BERT.
> > > >
> > > > On the other hand, the MemTransformer model studied here is *based on* a BERT-base model, so achieving a similar level of performance shouldn't be a huge surprise. Unlike the XLNet's "across-the-board-better" table that the authors point to above, it seems here that in this paper sometimes BERT is better, while at the other times MemTransformers (not just one model, but various ones with different # of memory slots) are better. I am therefore not fully convinced that whether the model truly improves over the baseline without further results, and whether the improvements has a relatively consistent pattern.
> > > >
> > > > And just to highlight, I completely agree with the authors that the numbers should not be what to pursue here; nor should it be the main focus of perhaps a lot of papers in the field. But it does not diminish the role of these numbers in conveying an important quality that we sometimes should look at when evaluating these papers: **how convincing are the methods, and how are they performing when compared to mature, well-developed, sufficiently large-scale baselines that this field has seen before?** I find these two questions very relevant to the goal that the authors are studying here (e.g., the benefit of adding memory slots).
> > > >
> > > > Thanks again for the clarification and I look forward to the updated draft.

---

### Author Response · Authors · 2020-11-25
**Thanks to all reviewers for their effort spent on discussion of our contribution**

We appreciate comments and suggestions from all reviewers and looking forward to use some ideas in further research. Unfortunately, we were not able to finish all our experiments to reproduce stronger baseline in time. Still, we believe that results already presented in the paper are valuable for a progress in the field and ask reviewers to reconsider scores.

---

### Decision · Program_Chairs · 2021-01-07
**Final Decision**

**Decision:**

Reject

**Comment:**

The paper studies three kinds of memory-augmented Transformers, focusing on one (the MemTransformer, which adds [MEM] tokens to a document.)  This is a nice clean extension of Transformers and a topic well worth investigating.  Unfortunately, the experimental results were considered unconvincing:

 - The baselines were relatively weak
 - The experimental setting was unusual (eg only 10 epochs)
 - The experiments did not show consistent improvement

Overall the paper was considered below acceptable quality for ICLR.